# Monitoring and Projecting Land Use/Land Cover Changes of Eleven Large Deltaic Areas in Greece from 1945 Onwards

**Anastasia Krina [1], Fotios Xystrakis [1], Kostas Karantininis [2] and Nikos Koutsias [1,*]**

[1] Department of Environmental Engineering, University of Patras, G. Seferi 2, GR-30100 Agrinio, Greece; anastasiakrina@gmail.com (A.K.); fotios.xystrakis@gmail.com (F.X.)
[2] Department of Work Science, Business Economics & Environmental Psychology (AEM), Swedish University of Agricultural Sciences (SLU), PO Box 88, SE-23053 Alnarp, Sweden; Karantininis.Konstantinos@slu.se
[*] Correspondence: nkoutsia@upatras.gr; Tel.: +30-26410-74201

**Abstract:** Wetlands are areas of high biodiversity and provide many ecosystem services of high value. However, they are under constant threat from intense anthropogenic pressures, mainly agriculture intensification, urbanization, pollution, and climate change. The temporal and spatial patterns of land use/land cover (LULC) changes within eleven large wetlands in Greece were analyzed based on thematic maps generated from aerial orthophotos taken in 1945, 1975, and 2007. Socio-economic developments and the consequent need for more arable land and utilization of water resources are among the factors that mainly determine their evolution. In 2007, LULC classes related to wetland vegetation were reduced to one third as compared to 1945 and they were mainly replaced with croplands and urban infrastructures. Each of the different sub-periods that was considered (1945–1975 and 1975–2007) was distinguished by characteristic patterns of change. Agricultural land increased up to 42% from 1945 to 1975 and became the dominant LULC class in all deltaic areas but Evros. A considerable stability was observed for the period 1975–2007 for all LULC classed but it is remarkable the extent of urban areas that doubled. There is a tendency of landscape simplification and homogenization among the deltaic areas and the output of Markov chain analysis indicates that future composition of deltaic landscapes will be similar to the current one if the main driving forces remain constant. Changes in LULC composition and structure are also combined with coastal erosion in all deltaic areas. This is attributed to the modification of sedimentary deposits due to dam construction. The results summarize the change trajectories of the major deltaic areas in Greece from 1945 to 2007 thus offering a great outlook of changes that allows managers to understand how policies and socio-economic requirements affect the deltaic ecosystems and what decisions should be made to protect and enhance them.

**Keywords:** wetlands; landscape metrics; object-based classification; aerial photos interpretation; Mediterranean; Markov chain; Monte Carlo randomization

## 1. Introduction

Wetlands are areas of high biodiversity and provide many highly valued ecosystem services [1–5], therefore awareness on the necessity of their conservation has grown over the last 50 years [6]. However, wetlands are under constant threat from intense anthropogenic pressures, such as intensification of agriculture, urbanization, pollution, and climate change [7]. The excessive exploitation of natural resources has led to a reduction in wetland surface areas, biodiversity loss, reduction of fresh water supplies, and rapid coastal erosion [8,9].

Human activities in the rivers and their catchment basins, such as deforestation, urbanization, and the expansion of crop cultivations have greatly changed the extent and nature of the materials that enter river estuaries. The conversion of forest land to almost any other land use promotes the surface flow of rainwater, increases the flow speed and magnitude of runoff, and increases sediment, organic matter, and the influx of inorganic nutrients [10,11]. Additionally, the likelihood of flooding in these areas also increases [12,13]. Coleman et al. [14] showed that in 20 years, 52.4% of plain land of fourteen deltaic areas has changed irreversibly, either due to natural causes or for agricultural or industrial purposes. In China, wetland area reduced by 33% from 1978 to 2008, mainly due to anthropogenic causes [15]. In Europe, wetland areas have reduced by one third from the beginning of the 20th century [16] and in Greece, the loss of wetlands from 1920 to 1991 has been estimated at 63% while the loss of riparian vegetation near to 35% [7].

Land use/land cover (LULC) changes and their implications have been studied extensively as shown by numerous available literature publications using different spatial data [17–24]. Agricultural and livestock practices, especially traditional ones, are considered two of the most important factors of change as a consequence either of the direct transformation of land for cultivation using fire [25], or as a practice to improve the land for livestock use [26–28].

LULC changes in Mediterranean coastlines and in coastal wetlands were investigated within the GlobWetland-II project [29] and the results showed that, in addition to agricultural intensification, urban sprawl and increasing demands for new tourist infrastructure were the most important land use amendment procedures that affect the reduction and continuous pollution of coastal wetland systems. Parcerisas et al. [30] concluded that the main cause of degradation of the landscape on the coast of Spain is the extension of urban fabric in combination with urban sprawl and construction projects. Alphan [31] identified a large number of changes in the coast of Turkey, including the transition of traditional crops to greenhouses and the expansion of crops at the expense of wetland ecosystems. Zuazo et al. [32] observed that the substitution of traditional crops with intensively irrigated crops further degraded a watershed in SE Spain with negative consequences on the sustainable use of resources. It has been estimated that just 4.7% of primary vegetation have remained unchanged in the Mediterranean and that the landscape changes continually [33,34].

Greek wetlands have been comprehensively studied by Zalidis et al. [35] and Papayannisand Pritchard [36] and both research teams agree that the factors most affecting Greek wetlands are: (i) the construction of irrigation systems and diversion of watercourses causing changes in the hydrological regime; (ii) land abstraction, land reclamation, and illegal hunting, all causing depletion of natural resources; (iii) pollution originating from agriculture and residential areas that causes a reduction of water quality; and (iv) increased construction and expansion of agricultural areas causing the loss of wetlands.

In this study we investigated the temporal qualitative and quantitative changes in the composition of LULC categories and landscape structure [37] in deltaic areas of Greece that have witnessed large-scale modifications during recent decades. The goal of the study was not only to present and analyze the synthesis and change of LULC classes in these areas, but to compare and classify them based on their characteristics and project their synthesis to the future. The study contributes to research on the degradation and loss of wetland ecosystems and provides a near complete picture of the current conservation status of deltaic areas at the landscape level in the eastern Mediterranean.

## 2. Study Areas

### 2.1. The Deltaic Areas

Eleven deltas that include almost all the large deltaic areas of Greece and are spatially located in the north, west, and center of the country were selected to study their LULC changes (Figure 1). Actually, these eleven deltaic areas include the estuaries of Greece's fifteen large rivers. All the deltas are subject to protection since they are included in the Natura 2000 network and the majority of them

are also classified as Wetlands of International Importance under the Ramsar Convention. All study areas have been studied individually in the past [38–42].

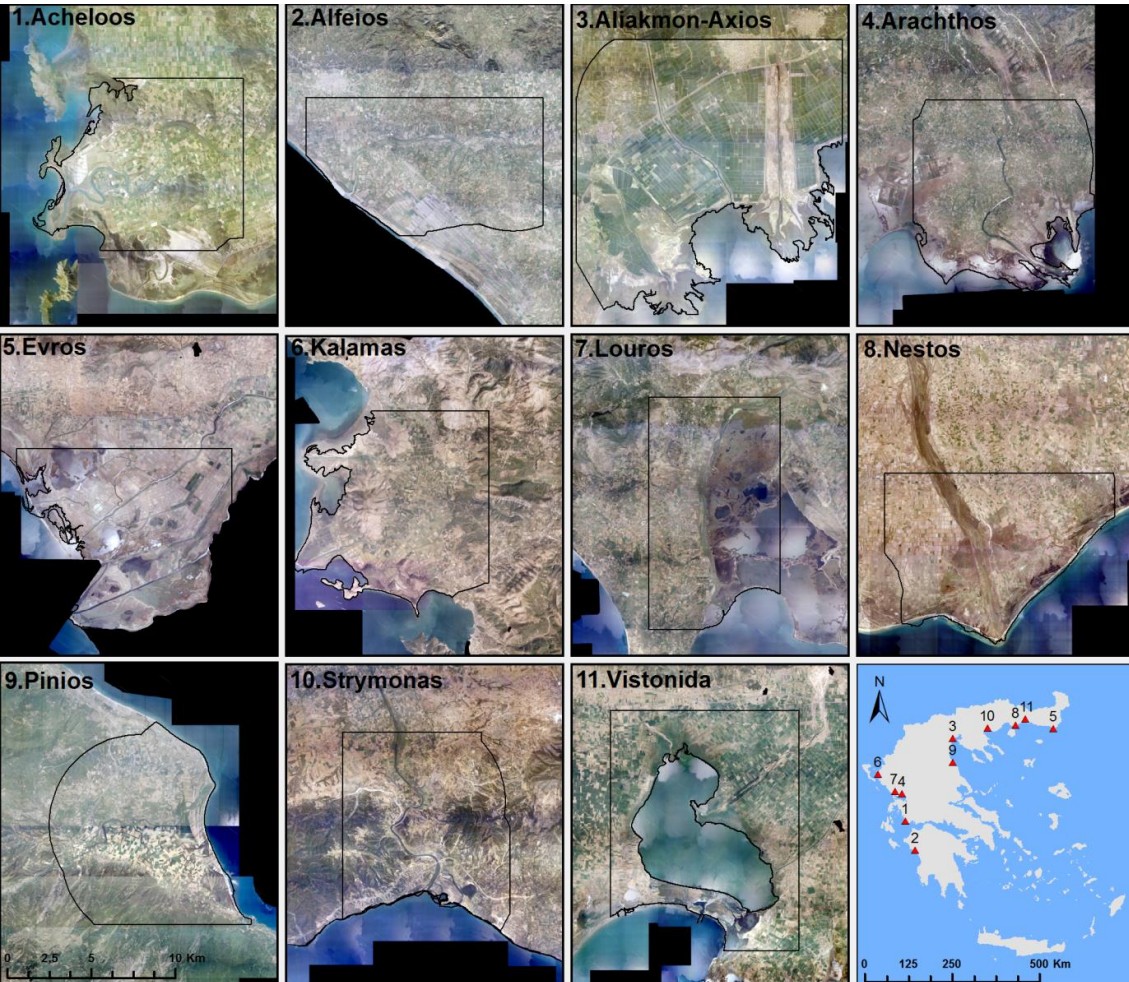

**Figure 1.** Geographical distribution and maps of the deltaic areas used in this study.

## 2.2. Definition of Deltaic Extents

Defining landscape extent is somewhat arbitrary and modifiable [43]. This is known in the literature as the modifiable areal unit problem (MAUP) and was first raised by Openshaw [44] as the fundamental geographic problem concerning the selection of study area limits and how these limits affect the results produced. This study does not address this problem, however, MAUP was taken into consideration when defining the extent of the deltaic areas. Recognizing the sensitivity of spatial analysis to the position and extent of the study areas, it is important to delineate the landscapes objectively based on the adoption of specific criteria (Figure 2). The adoption of these criteria guarantee for an objective assessment of the extent of the study areas. We did not define the deltaic extents (for example, based on the physical assessment of the drainage) to avoid creating study areas that would be very different in their extent. For example, the Aliakmon-Axios site would have been extended to several kilometers since it corresponds to the second biggest plateau in Greece.

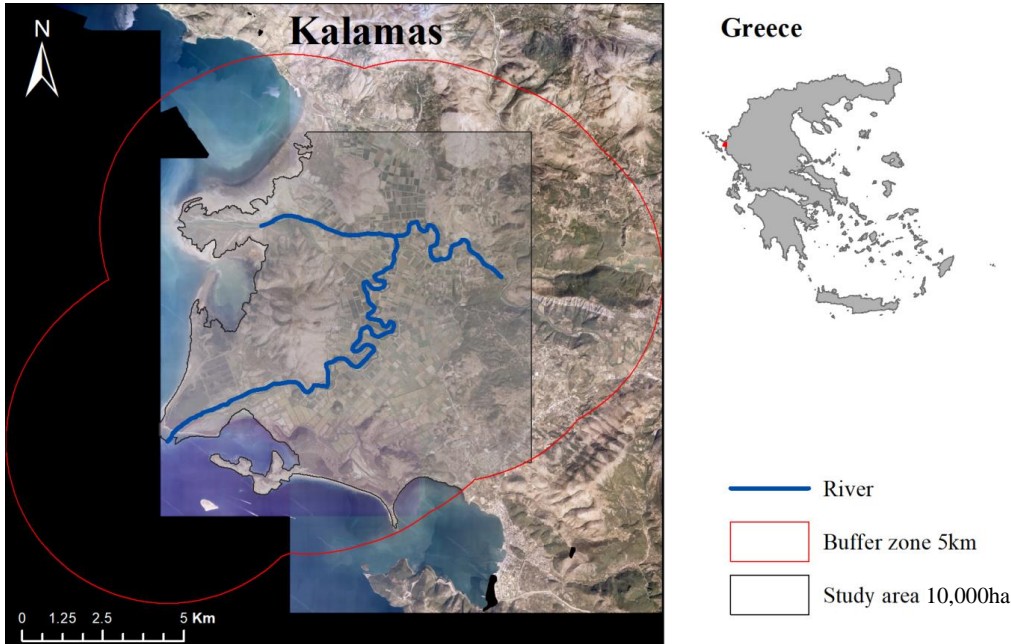

**Figure 2.** Schematic representation of the rules applied to define landscape extent.

Therefore, the border and extent of each delta was defined based on three rules: (i) each area has a size of 10,000 ha (100 km$^2$), with the exception of the Aliakmon-Axios site which has exactly double that size. It was considered that these two major river deltas interact and for this reason they were not separated into two separate study areas. (ii) The total area of each delta does not include the surface area covered by sea, unless at some point in time that particular area was land; and (iii) the third rule set concerns the shape of the study areas. Their width cannot not exceed a 5 km buffer on each side of the riverbed (Figure 2). The only exception to this rule is the Evros site where the River Evros forms the physical border between Greece and Turkey and thus it was not possible to equally extend the study area on both sides of the river.

## 3. Materials and Methods

### 3.1. Aerial and Satellite Photographs

The analysis of temporal and spatial patterns of LULC changes was based on thematic maps created from black and white and colored aerial orthophotos taken in the years 1945 (1:42,000) and 2007 (0.5 m resolution) that were provided by the National Cadastre and Mapping Agency S.A. For 1975 the land cover mapping was based on black and white aerial photographs provided by the United States Geological Survey (USGS). These images were part of a series of photographs taken for the KH-9 program, codenamed HEXAGON, where a series of photographic reconnaissance satellites were launched from the United States between 1973 and 1980. To allow compatibility with the first two datasets, these data were orthorectified and geo-referenced to the Greek Geodetic Reference System (EGSA 87) with a root mean square (RMS) error of less than 0.75 m for all cases (ranged from 0.48 m to 0.75 m for each case) compared to their ground-measured coordinates before further co-processing. Fiducials located in the corners of the oriented image were used for the interior orientation of the aerial photographs. Geometrical corrections were performed using ground control points defined in the orthophotos of 2007 and a digital elevation model (DEM) of 5 m resolution provided by the National Cadastre and the Mapping Agency S.A. The number of ground control points for each case ranged from 13 to 38 with a good spatial distribution over each photo. All images were spatially resampled to 4 m. This pixel size corresponds to the largest pixel size value among all available rasters (Figure 3).

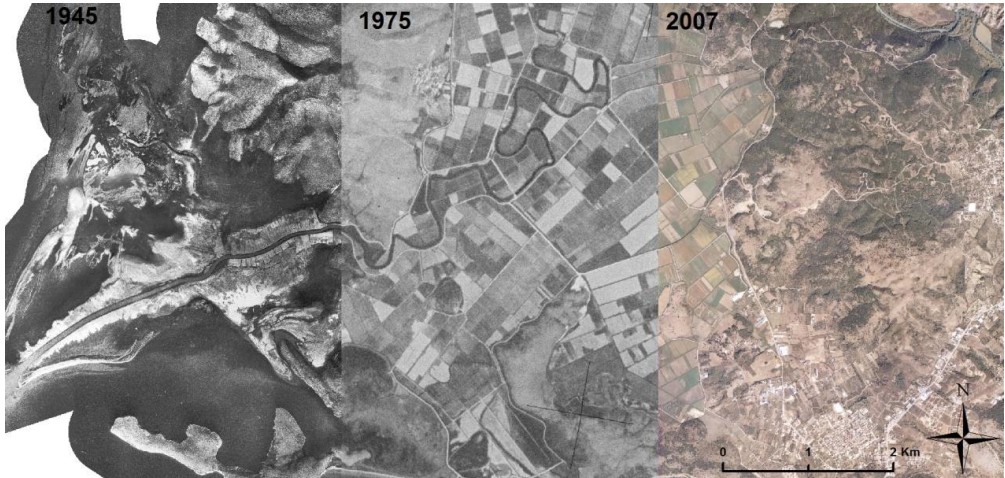

**Figure 3.** Series of aerial photographs (1945, 1975, 2007) used for the creation of thematic land use/land cover (LULC) maps and spatial matching following their geometric correction and adjustment. A section of Kalamas Delta is shown here as an example.

### 3.2. Land Use and Land Cover Classification Scheme

The classification system adopted in this study included generalized LULC classes that were modified to the particular characteristics of deltaic ecosystems. The level of detail mapped depends primarily on the resolution of the images used from the years 1945 and 1975, as the poor spectral resolution of grey-scaled aerial photos puts limitations on the detailed identification of LULC classes. To minimize the uncertainty of accurately mapping the LULC classes using a detailed classification scheme, we adopted a rather broad one that can guarantee a quite high accuracy in the mapping process. Therefore, the LULC classes included were: (1) urban areas, (2) agricultural areas, (3) wetlands, (4) inland water, (5) bare land, (6) grasslands, (7) shrublands, (8) forests, and (9) sea. Figure 4 shows an example of the classification scheme used in the series of geo-images from 1947, 1975, and 2007.

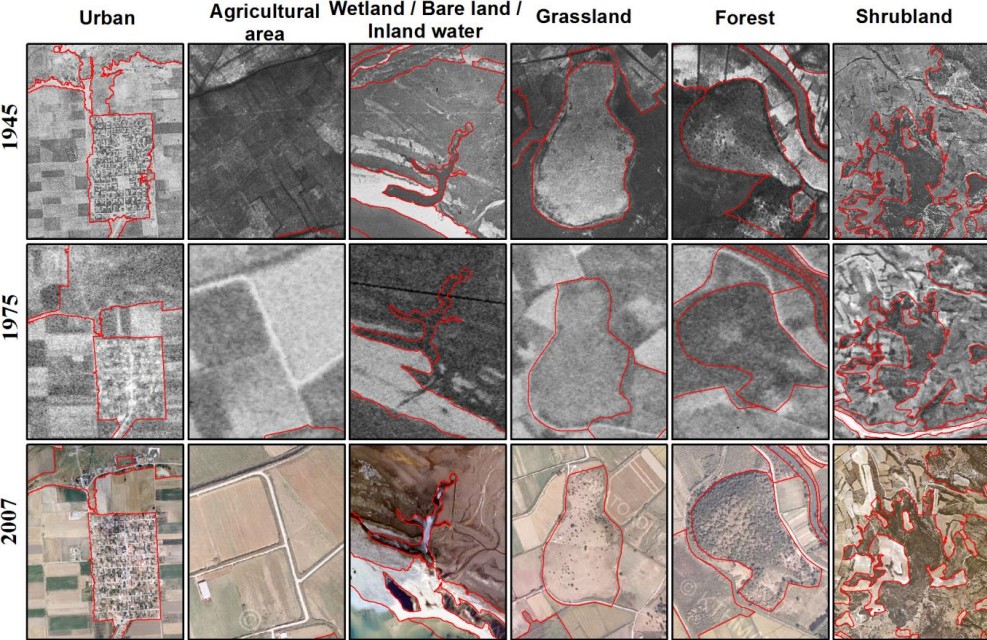

**Figure 4.** Graphical presentation of the LULC classification scheme in aerial photographs from 1945, 1975, and 2007. Specific photointerpretation keys were developed and used to create the thematic maps using the high-resolution aerial photographs together with the object-based classification approach.

### 3.3. Land Use and Land Cover Mapping

Orthophotos were initially processed by eCognition software [45]. The software produced the first vector layers to delineate the initial LULC classes from which the final thematic maps were generated. The algorithm used for the object-based segmentation of images was multiresolution segmentation. Specific values that are required for the segmentation process, including the scale, color, and shape parameters, were set independently for each deltaic area by means of visual inspection of the output. All images were segmented at two levels, one generalized and one detailed, which were created through the segmentation of the broad level. Both levels were processed in a GIS environment (ArcGIS) and manual polygon union took place until the desired output was achieved. The map of 2007, the one with the best image resolution and quality, was then used as background in order to assist the development of the LULC maps of 1975 and 1945, through the manual inspection and addition and deletion of segments from 1975 and 1945, respectively. Line smoothing at 20 m took place in order to reduce differences in object complexity among the produced maps that derive from the different pixel size of the original raster data. Finally, the LULC maps for each year were developed after the elimination of polygons less than 0.2 ha of the surface area. For the interpretation and classification of the segments of the produced maps, we also used the Corine thematic maps [46], habitat type maps of the Natura 2000 network [47] as well as various forest management maps.

### 3.4. Data Analysis

#### 3.4.1. Landscape Synthesis and Change

The landscape composition expressed by the number of pixels and total percent cover of each class was estimated per study site for the years 1945, 1975, and 2007 and the future conditions as estimated by the Markov equilibrium state. To describe the landscape composition and to quantify the spatio-temporal changes, cross-tabulation matrices were estimated for two sub-periods, that of 1945–1975 and 1975–2007. Additionally, the metrics proposed by Pontius et al. [48], including net change, swap change, and total change, were also estimated (Table 1).

**Table 1.** The cross-tabulation table structure.

| Time 2 | | | | Total Time 1 | Loss | Net Change (D) | Swap (S) | Total Change |
|---|---|---|---|---|---|---|---|---|
| Class 1 | Class 2 | Class 3 | Class 4 | | | | | |
| **Time 1** | | | | | | | | |
| Class 1 $P_{11}$ | $P_{12}$ | $P_{13}$ | $P_{14}$ | $P_{1+}$ | $P_{1+} - P_{11}$ | $abs(P_{1+}-P_{+1})$ | $2 \cdot min$ (gain, loss) | $D_1 + S_1$ |
| Class 2 $P_{21}$ | $P_{22}$ | $P_{23}$ | $P_{24}$ | $P_{2+}$ | $P_{2+} - P_{22}$ | $abs(P_{2+}-P_{+2})$ | $2 \cdot min$ (gain, loss) | $D_2 + S_2$ |
| Class 3 $P_{31}$ | $P_{32}$ | $P_{33}$ | $P_{34}$ | $P_{3+}$ | $P_{3+} - P_{33}$ | $abs(P_{3+}-P_{+3})$ | $2 \cdot min$ (gain, loss) | $D_3 + S_3$ |
| Class 4 $P_{41}$ | $P_{42}$ | $P_{43}$ | $P_{44}$ | $P_{4+}$ | $P_{4+} - P_{44}$ | $abs(P_{4+}-P_{+4})$ | $2 \cdot min$ (gain, loss) | $D_4 + S_4$ |
| Total time 2 $P_{+1}$ | $P_{+2}$ | $P_{+3}$ | $P_{+4}$ | 1 | | | | |
| Gain $P_{+1} - P_{11}$ | $P_{+2} - P_{22}$ | $P_{+3} - P_{33}$ | $P_{+4} - P_{44}$ | | | | | |

Rows represent the records of the first year (beginning of period) and columns represent the last year. Diagonal elements show the percentage of persistence of the class, and off-diagonal elements show the amount of transition between LULC classes during the time period considered. Totals of lines (Total time 1) and columns (Total time 2) illustrate the extent of each class at the beginning and end of each time period, respectively. The difference between the two summaries is the "Net change" of the applicable class. Net loss (Loss) and pure profit (Gain) were calculated for each class. The smallest of those two values doubled equals the total area of "Swap change" (Swap). The sum of "Net change" and "Swap change" gives the "Total Change" of each class.

Cluster analysis was used to determine if the study areas could be divided into groups with similar landscape composition per time period. The partitioning around medoids (PAM) method [49] identifies representative objects (medoids) in each cluster (i.e., a mathematical representative object with the lowest average dissimilarity of all other objects in the group). It is considered a robust clustering

method that is less sensitive to outliers [49,50]. Additionally, this method allows for silhouette graphical representation [51] from which the optimal number of groups and their degree of consistency can be easily identified [49]. Euclidean distance was used as the distance measure and the nature of the data (percentages) did not require standardization [52]. The analysis was performed using the 'fpc' [53] and the 'cluster' [54] R-packages. As the results of the clustering were considered to have a weak structure based on the silhouette coefficient, hierarchical cluster analysis using Ward's linkage method and Euclidean distance was performed in R [55] to compare the outputs.

### 3.4.2. Landscape Structure and Change

Landscape metrics are sensitive to scale, including the resolution of the thematic data [56–59]. Therefore, landscape analysis was not performed for the year 1975, as the resolution (scale) of these images was much lower than the other data. This difference would not allow homogeneous data processing or direct comparison of results. Landscape metrics were calculated using raster format data of 5 m resolution in FRAGSTATS 4.1 [60]. The choice of 5 m resolution was based on the comparison of the outputs of sequential analyses that were performed using input rasters of 2 to 7 m pixel size (pixel size increased by 1 m in every sequential step). The choice of different values of pixel size did not qualitatively alter the outputs, but the choice of 5 m pixel size provided slightly better results in terms of significance and correlation strength when the landscape metrics were used in consecutive analyses. Initially, 28 metrics were calculated at landscape level and only those with correlation coefficients less than 0.9 were retained, since in many cases the indices were redundant as they are highly correlated with each other [61–63]. The final landscape metrics used to characterize the landscape structure and study aspects of landscape change were: PD (patch density), LPI (large patch index), SIEI (Simpson's evenness index), SHDI (Shannon's diversity index), GYRATE_AM (area weighted mean radius of gyration), and FRAC_AM (area weighted mean patch fractal dimension).

### 3.4.3. Patterns of Change

A Monte Carlo randomization test [64] was applied to investigate the pattern of the LULC changes occurring from 1945 to 2007 (e.g., random pattern). The changes observed during the period 1945–2007 were spatially randomized one thousand times over the thematic map of 1945 (initial landscape) to identify selective patterns (preference or avoidance) of changes over specific landscape classes. If all LULC classes were equally likely to change, the change ratio would be identical to the percentage of the area occupied by each class in the initial landscape. Any statistically significant variation from this ratio would signify selectivity (preference or avoidance) of change for specific LULC classes [65]. To generalize the results the deltaic areas were grouped based on the outputs of the cluster analysis and then we calculated the within group summary of (a) the total area of the available landscape (1945) LULC classes, and (b) the total area of the observed changes. We then examined patterns of changes and possible selectivity indications at the group level.

### 3.4.4. Markov Analysis

LULC distribution was assumed to follow a stationary first order Markovian process [66]. Thus, the distribution of the various LULC categories in each delta at time t, depends on their distribution at the previous time t − 1, and a transition probability matrix (TPM) maps the distribution t − 1 onto t. The objective is to recover the TPM from a set of limited macro data on distributions. In this study the distributions of nine LULC categories were examined over three years (1947, 1975, 2007), hence there are two transitions for each delta and a total of $3 \times 9 = 27$ data points. A TPM of $9 \times 9 = 81$ elements was estimated (recovered). This is a classic case of an ill-posed problem. While several methods of recovering the TPM from limited macro data have been developed [66], the generalized maximum entropy (ME) formalism was selected for use in this study. The ME was developed by Lee and Judge [67] and Golan et al. [68], and extended by Karantininis [69] (for further details see the

original publications). Additionally, the use of generalized cross entropy (GCE) [67,68] allowed the incorporation of prior information, for example, that derived from aggregate data.

Since the Markov model illustrates a dynamic process of the propagation of a distribution of "states" over time, it is useful to determine whether the system tends towards equilibrium and what this equilibrium is. In other words, is the matrix ergodic? Does it tend to equilibrium regardless of the initial state? "The equilibrium distribution of interest [is] not a forecast of what the future state of the industry will be but as a projection of what it would be if the observed pattern of movement continued. It is thus an indication of the tendencies at work within the distribution." [70], p. 306. The distribution of a regular Markov chain is ergodic if it converges to a unique invariant distribution. This equilibrium distribution is shown to be the eigenvector of the TPM [71].

The CONOPT algorithm of the General Algebraic Modeling System (GAMS) was applied in this study to recover the TPM for each delta. GCE formalism was employed using the recovered TPM for the country aggregate of deltas as priors. That is, firstly the hectares for each LULC category were aggregated for the entire country, and then the aggregate TPM was recovered using ME. Each element of the recovered TPM aggregate was used as the prior to recover the TPM for each delta. Equilibria were computed for each delta by calculating the eigenvector for each recovered TPM.

## 4. Results

All the results of this study, including the thematic maps of the years 1945, 1975, and 2007, the thematic and spatial changes recorded in 1945–1975 and 1975–2007, landscape synthesis and structure, selectivity analysis, and the transition matrices for each delta, are presented in the form of tables and graphs in the Supplementary Material. The complete data set produced for Kalamas Delta is presented in Figure 5 as an example.

### 4.1. Change Detection

The transition matrices (see Supplementary Material) revealed the extent of changes that took place over the examined time periods in all eleven deltaic areas. The first period of 1945–1975 was characterized by rapid changes in land use, the massive loss of natural areas, and a simultaneous increase in cultivated areas in all deltaic areas. The LULC classes prevailing in 1945 were 'agricultural area' and 'wetlands', while non-natural areas, including construction and built-up areas, represented an average of just 0.53% in all areas. Cultivations occupied an average area of 36.34% (ranging from 0.21% to 59.36%). The area occupied by wetlands varied between the study sites and ranged from 6.12% to 62.82%. The proportions of wetlands observed in Pineios (6.12%) and Strymonas (7.69%) sites can be roughly characterized as 'outliers' when compared to the other deltaic areas. The small extent of wetlands recorded in these two sites is compensated by their high percentages of forest and shrubland cover that reached 30.58% and 37.74%, respectively. In contrast, in 1945, Evros was mainly characterized by its large wetland that covered 62.82% of the site.

The LULC changes that occurred in the period from 1945 to 1975 revealed major transformations in the landscape and a distinct trend of homogenization between the different landscape areas in all sites. This was mainly due to the observed increase in agricultural land, which ranged from 3.18% to 42.09%. Agricultural land became the dominant LULC class in all regions with the exception of Evros where 'wetlands' dominated. The swap changes in the sites varied from 0.30% to 7.88%. Changes to the coastline were also significant during this time period, as the area of polygons representing the sea decreased in all sites but Alfeios. The difference between gain and loss of sea, as shown in the transition matrix in Figure 5 and the Supplementary Material, indicates that increase in sea surface ranged from 0% in Alfeios, to −4.39% in Aliakmona-Axios, with an average value of −1.55% over all deltaic areas. It is interesting to observe that not a single deltaic area has a positive value for the net change in the surface of the sea.

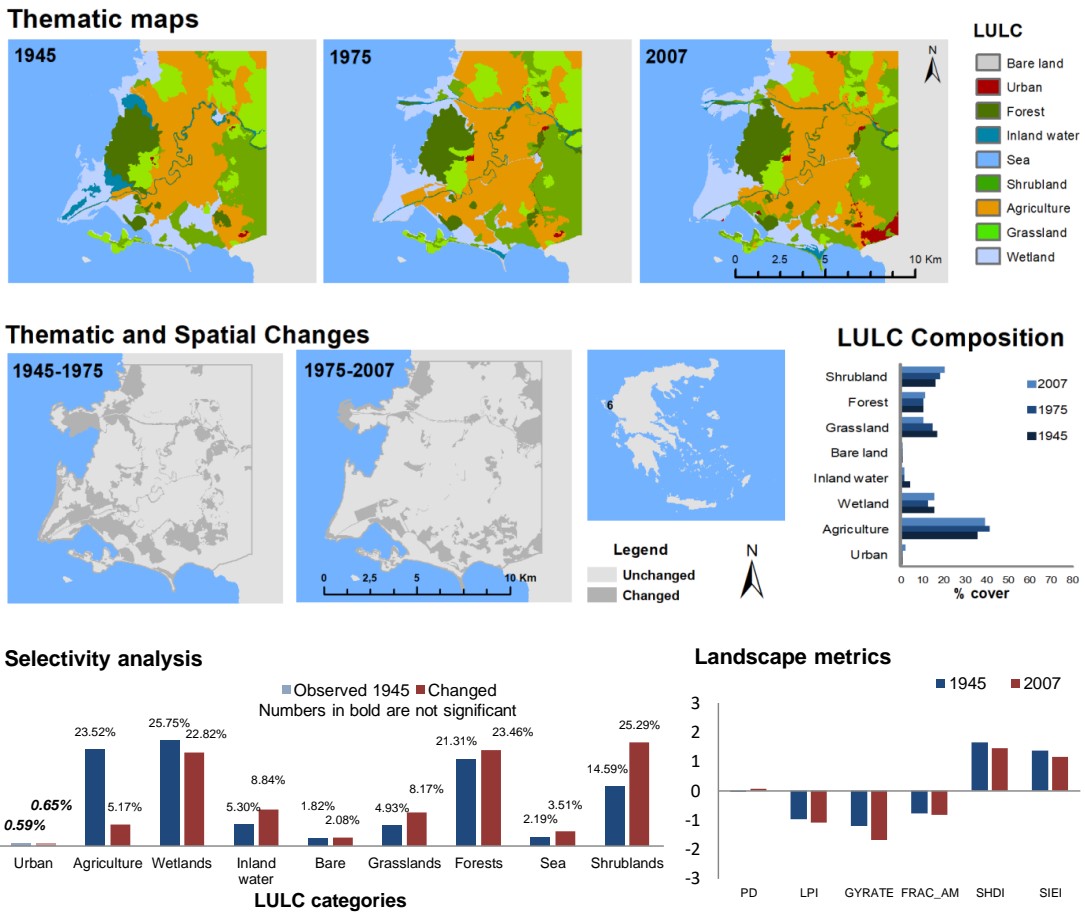

**Figure 5.** Deltaic area-based principle outputs: LULC thematic maps, changes, LULC composition, selectivity of changes, landscape metrics, and transition matrices. The example of Kalamas Delta is shown here. Data for all the deltas studied are presented in the Supplementary Material.

From 1975 to 2007, the intensity of LULC change decreased in all study areas. The percentage of land that did not change ranged from 74.16% to 91.82%. However, an inverse trend of land use was also observed as the surface of cultivated land decreased in four of the eleven study sites, the surfaces of

forests increased in seven sites, and eight deltaic areas experienced coastal erosion. In 2007, the extent of urban areas had almost doubled compared to 1975, reaching a mean value of 2.01%, while the mean area of cultivated land ranged from 21.36% to 78.36%. Moreover, during this period, the surface area covered by the sea increased indicating that coastal erosion takes place (Figure 5 and Supplementary Material). Nevertheless, this is not uniform for all deltaic areas: In Arachtos, Kalamas, and Vistonida an overall decrease in the area surface of sea is observed. The largest increase is sea surface is observed in Aliakmonas (1.35%), while the largest decrease is in Vistonida (−1.47%). In average, a gain of 0.1% in sea surface is observed over all deltaic areas.

*4.2. Cluster Analysis*

Results of the cluster analysis are presented in Figure 6 According to the empirical coefficients proposed by Kaufman and Rousseeuw [51], the values of the consistency index were low for each time period, therefore, the group discrimination was verified by additional methods. The outputs of the hierarchical clustering (Figure 6) verified the groups defined by the PAM method in each time period (1945–1975 and 1975–2007).

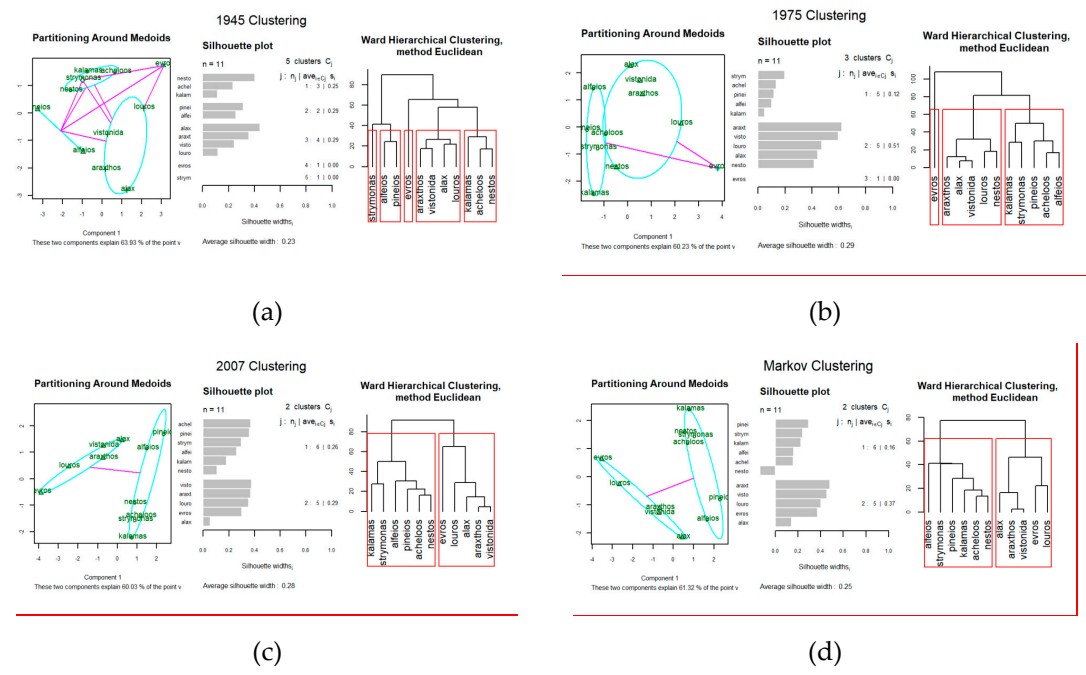

**Figure 6.** Cluster analysis outputs of landscape composition for the years (**a**) 1945, (**b**) 1975, (**c**) 2007 and (**d**) for future projection. The first two columns refer to the non-hierarchical partitioning around medoids (PAM) method. The third column shows the outputs of hierarchical clustering. Labels of deltaic areas when abbreviated: achel: Acheloos; alax: Aliakmonas-Axios; alfei: Alfeios; araxt: Arachthos; evros: Evros; louro Louros; nesto: Nestos; pinei: Pineios; strymo: Strymonas; visto: Vistonida.

The analysis shows that the composition of LULC classes in the deltaic areas tended towards homogenization. The five groups recorded in 1945 gradually decreased to three in 1975 and two in 2007. In 1945 the Evros river delta presented a distinct composition defined by its large surface covered by wetlands. However, in 2007, Evros grouped with the deltas of Louros, Aliakmona-Axios, Arachthos, and Vistonida that also comprise extensive wetland vegetation (Figure 7). The discrimination of the two groups observed in 2007 is based mainly on the differences in the composition of the (semi-)natural, terrestrial LULC classes. The deltaic areas of group 1 are characterized by high cover of grasslands, shrublands, and forests (Figure 7). If the contemporary grouping of 2007 is accepted (i.e., a distinct division of the deltaic areas into two groups) and applied for the years 1945 and 1975, the among-group

differences in LULC composition reflect a tendency to separate the deltaic landscapes into "more wetland (group 2)" or "more terrestrial (group 1)" ecosystems already from 1945 (Figure 7).

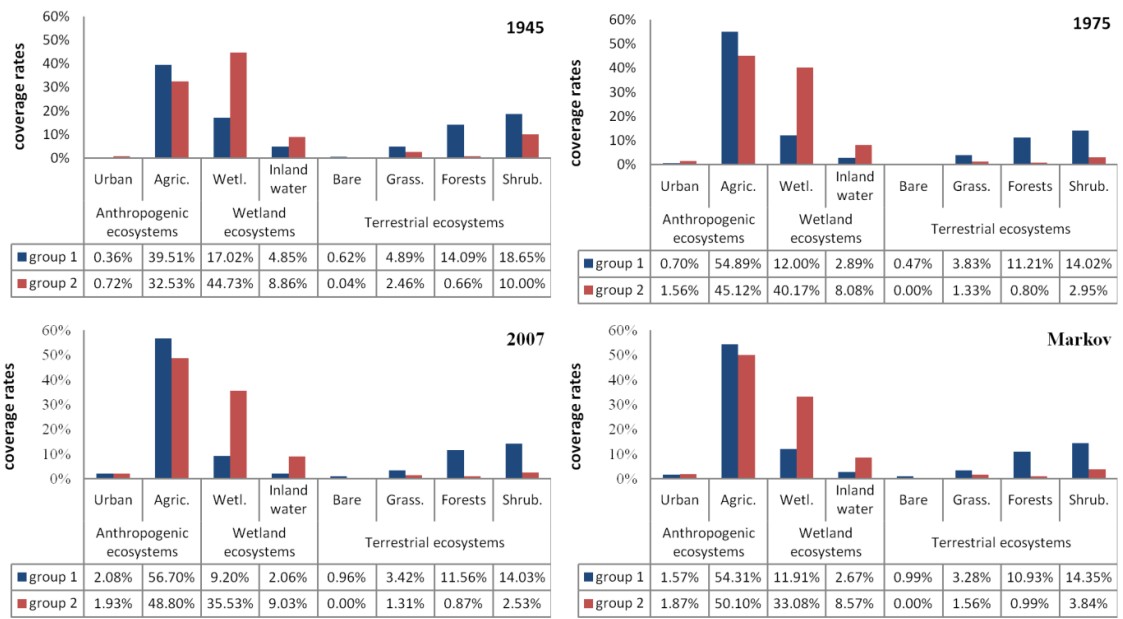

**Figure 7.** Averaged landscape composition of group 1 (blue) and group 2 (red) for the years 1945, 1975, and 2007, and the expected Markov equilibrium state. Agric.: agricultural land; Wetl.: wetlands, Grass.: grasslands; Shrub.: shrublands; Bare: bare land.

In 1945 anthropogenic ecosystems occupied areas of about 40% and 33%, wetlands occupied 22% and 53%, and terrestrial ecosystems occupied 38% and 13% in groups 1 and 2, respectively. Similarly, in 1975 the averaged composition of the two groups comprised 56% and 47% anthropogenic ecosystems, 15% and 48% wetlands, and 30% and 5% of terrestrial ecosystems in groups 1 and 2, respectively. Finally, in 2007 about 50% of the area of both groups were occupied by anthropogenic ecosystems, wetlands covered 12% and 45%, while terrestrial ecosystems covered 41% and 5% in groups 1 and 2, respectively. The Markov analysis outputs (Figure 7) indicated that if LULC changes are driven from the same forces and with the same intensity, the deltaic areas would be eventually clustered into two groups in the future (Figure 6), both having similar LULC syntheses as the groups recorded from the 2007 data (Figure 7). The first group will be largely characterized by the high percentage of terrestrial ecosystems and the second by wetlands. In both groups, however, agricultural land will dominate the future landscape.

## 4.3. Landscape Structure

Table 2 presents the values of structure metrics for each region and year studied. In most of the study areas the FRAC_AM, a measure of shape complexity, is seen to decrease over the years, thus highlighting the trend of landscapes to become less complex. This can be explained to a certain degree by the replacement of natural areas (which usually have irregular shapes) with agricultural areas that exhibit much less shape complexity. Additionally, the Simpson's evenness index, which considers the evenness of patch sizes, is seen to decrease in almost all study areas, indicating the dominance of certain LULC classes.

**Table 2.** Landscape structure metrics for each delta and year studied. PD: patch density; LPI: large patch index; GRYRATE_AM: area weighted mean radius of gyration; FRAC_AM: area weighted mean patch fractal dimension; SHDI: Shannon's diversity index; SIEI: Simpson's evenness index.

| Site | Year | PD | LPI | GYRATE_AM | FRAC_AM | SHDI | SIEI |
|---|---|---|---|---|---|---|---|
| Acheloos | 1945 | 0.90 | 17.79 | 1578.6 | 1.17 | 1.55 | 0.91 |
|  | 2007 | 0.92 | 31.31 | 2043.88 | 1.15 | 1.28 | 0.67 |
| Alfeios | 1945 | 2.41 | 24.19 | 1753.73 | 1.20 | 1.32 | 0.71 |
|  | 2007 | 2.14 | 38.83 | 2184.64 | 1.21 | 0.84 | 0.43 |
| Aliakmona-Axios | 1945 | 0.45 | 10.54 | 1770.54 | 1.16 | 0.96 | 0.76 |
|  | 2007 | 0.69 | 15.35 | 1537.92 | 1.11 | 0.99 | 0.59 |
| Arachthos | 1945 | 0.62 | 26.15 | 1769.58 | 1.17 | 1.11 | 0.69 |
|  | 2007 | 0.95 | 27.66 | 1972.56 | 1.20 | 1.17 | 0.69 |
| Evros | 1945 | 0.61 | 38.83 | 1913.18 | 1.19 | 1.01 | 0.65 |
|  | 2007 | 0.74 | 18.13 | 1632.93 | 1.17 | 1.09 | 0.69 |
| Kalamas | 1945 | 1.26 | 15.07 | 1309.39 | 1.16 | 1.65 | 0.89 |
|  | 2007 | 1.33 | 14.11 | 1144.04 | 1.15 | 1.60 | 0.87 |
| Louros | 1945 | 0.99 | 30.82 | 2213.00 | 1.18 | 1.50 | 0.86 |
|  | 2007 | 0.76 | 39.14 | 2539.12 | 1.1741 | 1.33 | 0.80 |
| Nestos | 1945 | 1.07 | 11.47 | 1344.56 | 1.19 | 1.72 | 0.91 |
|  | 2007 | 0.90 | 26.94 | 1627.63 | 1.16 | 1.34 | 0.72 |
| Pineios | 1945 | 2.47 | 27.90 | 1724.50 | 1.24 | 1.36 | 0.77 |
|  | 2007 | 1.97 | 24.28 | 1675.74 | 1.20 | 1.27 | 0.72 |
| Strymonas | 1945 | 2.12 | 15.98 | 1312.32 | 1.22 | 1.09 | 0.70 |
|  | 2007 | 2.90 | 15.5 | 1297.80 | 1.21 | 1.13 | 0.69 |
| Vistonida | 1945 | 1.35 | 25.98 | 1694.58 | 1.19 | 1.26 | 0.79 |
|  | 2007 | 0.59 | 22.39 | 1629.55 | 1.14 | 1.03 | 0.65 |

The correlation coefficients between the structure metrics and the LULC composition of all deltaic areas overall years are presented in Table 3. The observed negative correlation between wetland and patch density (PD) implies that the higher the percentage of wetlands in a deltaic area, the less fragmented the landscape. The positive correlation between 'inland waters' and the 'weighted average of radius of gyration' (GYRATE), which is a measure of elongation of patches, was expected since the class 'inland water' refers to rivers and lakes within the deltaic areas. Finally, grasslands showed a positive correlation with Shannon's diversity index (SHDI) and Simpson's evenness index (SIEI).

**Table 3.** Spearman's correlation coefficients between landscape composition and structure metrics. ** and * denote significant correlation at the 0.01 and 0.05 level, respectively.

| Categories | PD | LPI | GYRATE | FRAC_AM | SHDI | SIEI |
|---|---|---|---|---|---|---|
| Urban | 0.0536 | 0.2034 | 0.0444 | −0.0088 | 0.1996 | 0.1270 |
| Agriculture | 0.4089 | 0.0549 | 0.0372 | 0.1626 | 0.3177 | 0.4734 * |
| Wetland | −0.7750 ** | 0.0890 | 0.3205 | −0.4273* | 0.2477 | 0.0090 |
| Inland water | 0.4746 * | 0.3832 | 0.6240 ** | −0.1955 | 0.0086 | 0.1604 |
| Bare land | 0.0991 | 0.0023 | −0.1167 | 0.1535 | 0.3637 | 0.1898 |
| Grassland | 0.0273 | 0.2635 | −0.3992 | −0.2834 | 0.6787 ** | 0.5585 ** |
| Forest | 0.3590 | 0.0466 | −0.2143 | 0.3260 | 0.5168 * | 0.3075 |
| Shrubland | 0.5142 * | 0.2183 | −0.4920 * | 0.3280 | 0.1525 | 0.1206 |

*4.4. Patterns of Change*

Selectivity analysis revealed that changes occurring within the period 1945–2007 were very selective (either positive or negative) to the LULC classes of 1945 (see Supplementary Material). In other words, due to different selective mechanisms the changes observed during this period refer to specific LULC classes. For example, with the exception of the Evros Delta where agricultural areas are mostly absent, the LULC changes in all other deltaic areas show a significant negative selectivity with

respect to the agricultural areas recorded in 1945. In the deltaic areas of Alfeios, Aliakmona-Axios, Nestos, Louros, Pineios, and Strymonas, LULC changes are significantly preferential with respect to 'wetlands'. In all the deltaic areas studied, LULC changes are preferential (positive) to 'grasslands' and 'bare land', whereas in the deltaic areas of Acheloos, Kalamas, and Pineios there is a significant avoidance in the LULC class 'forests'.

In addition, a general pattern can be observed when the deltaic areas are grouped on the basis of the cluster analysis (Figure 8). Deltaic areas of group 1 are characterized by a distinct positive selectivity of changes in wetlands, while in group 2 this pattern is not as prominent. In contrast, the distinct positive selectivity of changes in LULC class 'shrublands' observed in group 2, is not prominent in deltaic areas belonging to group 1. Finally, inverse patterns of selectivity are observed in the inland waters; in the deltaic areas of group 1 selectivity has a positive sign, while in group 2 it has a negative sign.

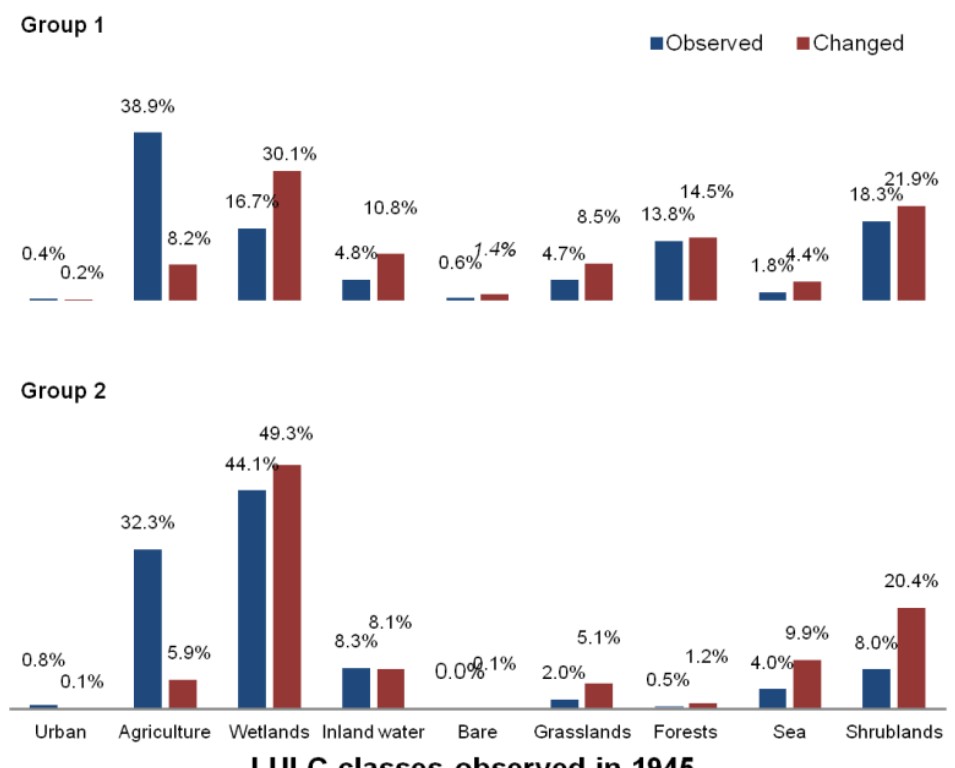

**Figure 8.** Landscape observed in 1945 (blue) and changes (red) that occurred from 1945 to 2007 for each LULC class in the deltaic areas of groups 1 (top) and 2 (bottom). When changes exceed the observed (initial) landscape this is an indication of positive selectivity (preference), while the opposite indicates negative selectivity (avoidance). Group-wise observed landscape composition and changes are calculated as the summary of the respective observed landscape composition and changes of the deltaic areas belonging to each group.

### 4.5. Future Projections

Comparison between the eigenvectors of the Markov TPMs and the last observed distributions (of 2007), suggests that all the deltaic areas studied have almost reached their steady-state and few changes are expected in the future (Figures 6 and 7). This is true if one assumes that the extent and intensity of the factors that triggered the recent changes remain constant. The projected future landscape allows the deltaic areas to be grouped into two distinct clusters with almost the same composition as the 2007 landscape (Figure 6). However, some changes are observed. For example, the Evros Delta that in 1945 presented a distinct landscape composition and was only grouped with other deltas in 1975, is grouped with the delta of Louros, as it now differs less from the other sites. A slight

decrease in agricultural land and forest cover is expected to occur in the sites of group 1, although this will be mainly compensated for by an increase in wetlands and in inland waters (Figure 7). A somewhat converse pattern is expected to be seen in the sites that make up group 2. Here, agricultural land and shrublands are expected to increase their cover while wetlands are expected to decrease (Figure 7). It should be noted that the LULC class 'urban' decreases in both groups of sites (Figure 7) according to future projections.

## 5. Discussion

This study demonstrated that although each deltaic area studied followed different patterns of landscape changes, some trajectories common to all sites can be distinguished. The synthesis of the landscape in the major deltas of Greece has gradually homogenized and become dominated by LULC classes that are related to human activity. It should be noted, however, that the results of this study were based on just ten LULC classes thus leading to a somewhat simplified categorization of the deltas, especially when considering the great heterogeneity of the vegetation occurring in deltaic regions [72]. It has been shown that landscape change trajectories can differ even between adjacent sites, depending on a number of ecological and social factors that act on a local scale [73]. Site-specific socio-economic background can determine the evolution of the landscape of each delta studied. For example, in 1945 the deltaic area of Nestos was grouped together with other sites that were characterized by high percentages of terrestrial ecosystem cover. However, during the 1950s the Greek state passed a decision to clear-cut the extensive riparian forest of "Kotza Orman" and release the area for agriculture. This explains why in 1975, Nestos Delta was grouped with deltaic areas that were characterized by a relatively high cover of wetlands. Nevertheless, output generalization allows for the identification of 'global' patterns that can provide insights suitable for the elaboration of national-scale strategic planning.

In Europe, socio-economic factors and implemented agricultural policies supporting intensification of production have led to a generalized trend of reduced landscape heterogeneity [74]. This pattern can also be seen in the results of this study. The variability at landscape level reflected by the large number of groups observed in 1945, gradually reduced to two groups of deltaic areas in 2007, each with distinct features that seem to be stabilized in the future. Agricultural land clearly dominates the landscape in both groups, yet the ratio of wetland to terrestrial ecosystems varies significantly between them. Agricultural intensification is accompanied by a decrease in landscape diversity [75] as agricultural land cover steadily increases at the expense of other, natural, LULC classes. Landscape complexity is often related to human interventions [76,77] but our results suggest that the intensity of agricultural intensification is an actor of landscape homogeneity. In the Iberian Peninsula, agricultural intensification was a major threat for small-scale wetlands inducing a decline of up to 94% in the area of wetland ecosystems since the beginning of the last century [78]. The intensification of irrigated crop farming contributed to the homogeneity of the landscape and water scarcity in coastal agricultural plains [79].

Furthermore, agricultural intensification and land use change at the catchment level can be a threat for a number of protected habitat types of the 92/43/EEC Directive [80,81] and is a major threat to biodiversity as it also leads to biotic homogenization [82,83]. This is why criteria like the assessment of the surface area and the range (= potential distribution) covered by the habitat types, as well as a rough estimation of the observed changes of the area covered by each habitat type at the national scale, are included in the Article 17 reporting format of the 92/43/EEC Directive [84]. Moreover, in the Standard Data Form, information regarding the human activities in each Natura 2000 protected area should be reported, among others, in the form of the surface area of the site affected by human activity [85]. At the moment, there is not a clear description regarding the methods to quantify of all of the aforementioned assessment criteria [86,87]. Their vague definition forces them to be currently treated as qualitative proxies assisting evaluators in the assessment of the conservation status of habitat types. The output of our (or other similar) study can be used for the quantification and the assessment

of such criteria as it provides homogeneous estimations of landscape composition and LULC changes of all major Natura 2000 wetland sites at fine spatial scale. Of course, the correspondence between LULC classes and habitat types of the Habitats Directive should be elaborated.

The LULC class that experienced the greatest reduction in surface area is wetlands. This characteristic land cover class of deltaic areas has been dramatically reduced and in 2007 wetlands covered only a third of the surface they covered in 1945. This figure is in agreement with Zalidis, Mantzavelas, and Gourvelou [35] who concluded that the expansion of infrastructure and agriculture are responsible for the loss of wetlands, affecting 52% of the estuaries studied. Similarly, the Commission of the European Community [16] notes that losses of wetlands from 1920 to 1991 were estimated at 63%, while Smardon [7] states that in the last two decades 35% of riparian vegetation alone has been lost. This decrease in wetland ecosystems has been accompanied by an increase in agricultural land or anthropogenic LULC classes in general, as shown not only from the results of this study, but also from numerous similar studies conducted in Greece, the Mediterranean, and worldwide [14,29,31,35,88,89].

Several LULC changes are worth special mention, not due to their spatial extent but due to the impact they have; these include the expansion of urban networks and coastal erosion. In Greece, where 90% of the population lives along the coastline [90], the growth of settlements in deltaic areas was mostly anticipated after 1989. According to Alexandridis et al. [91], from 1989–2001 the Greek deltaic region of Axios-Loudias-Aliakmona was subjected to a 27% increase in settlement growth, an 8% increase in road extension and widening, and a population growth of 11%. Similar patterns, especially in the coastal zone, were also observed in Acheloos Delta, in which touristic exploitation altered a significant proportion of the coastline [41]. The growth of urban and industrial LULC classes, despite their small footprint in terms of area covered, signifies disproportionally higher pressures on wetland ecosystems [92]. According to Gibbs [93], as the density of human activity increases, the density of wetlands lessens, and the their isolation increases.

Coastal erosion in deltaic areas is related to large-scale dam construction, river-route alteration, and the development of touristic infrastructure on the shoreline [8,41,42]. Deltaic ecosystems and estuaries are dynamic sedimentary systems sited between land and marine systems, therefore their form and functions are simultaneously affected by river and marine processes [94]. Dam building and river boxing alter sediment flow and modify the dynamic equilibrium of deltas, increasing the danger of coastal erosion [8,95]. Similar to other major deltas around the globe [96,97], in eight of the eleven deltaic areas studied here, coastal erosion accelerated following dam construction. In the delta of Axios-Loudias-Aliakmona, excessive use of natural resources led not only to a reduction in wetland surface and biodiversity loss, but also to a reduction in the supply of fresh water that subsequently led to rapid coastal erosion [8,9]. In this area, Axios' and Aliakmonas' discharges have been drastically reduced due to regulation of water flow by a large number of dams leading to a subsequent reduction in the deposition dynamic [98]. The effects of dam construction on coastlines are also apparent in other deltaic areas in Greece, such as Nestos [39,99], Strymonas [39], Alfeios, Arachthos, Louros, Sperchios and Inoi [40] and Axios-Aliakmonas [98]. Furthermore, it should be noted that the effects of dam construction are characterized by a delay in the appearances of their effect. For example, large-scale interventions that took place during the early post-WWII years led to coastal erosion that became evident after 1975.

The gradual changes in landscape composition confirmed by the results of this study largely reflect the socio-economic changes of modern, post-war, Greek society. Directly following WWII, many projects were carried out with the aim of expanding productive soils for agriculture. Major wetland drainage projects and dam construction were frequently financed in Greece [88], resulting in the rapid increase of arable land in all deltaic areas. In addition, the improvement of mechanical means for agricultural production aided the intensification of agriculture in the most accessible and productive sites. Wetlands, of course, are considered among these productive sites and were frequently targeted during these extensive land reformation projects.

The years 1975–2007 can be divided into two distinct parts regarding socio-economic conditions in Greece and Europe. Early in this period, extensive public works were completed, however some local drainage and regulatory dam work was still taking place in several deltaic areas (e.g., Evros and Nestos). Agricultural intensification was still observed and LULC changes occurred but not at the same rate as from 1945 to 1975. A distinct characteristic of the early decades of 1975–2007 period is that, gradually, agricultural intensification and abandonment are seen to co-exist in different areas within the same region [100–102]. Agricultural intensification and mechanization still take place in productive sites while cropland abandonment and the consecutive secondary succession of vegetation [103,104] is now distinct in inaccessible, mountainous sites [79,101,105,106] mostly as a result of the implementation of agricultural policies and financial development. This signifies the transition to the later, last few decades of 1975–2007, that is defined by increasing social awareness regarding the high aesthetic, cultural, and biodiverse value of wetlands. Most of the deltaic areas studied here are included in the Ramsar Convention [107] and the Natura 2000 network. More comprehensive nature conservation planning is proposed, although further steps still remain to be taken [108]. Large-scale interventions, likely to reduce the conservation status of habitat types or threatened populations of protected animal or plant species, are restricted and this explains the decrease in the intensity of changes towards anthropogenic LULC classes and the respective increase in the cover of natural LULC classes observed. During the last examined period (1975–2007), in most of the deltaic areas studied, the surface cover of agricultural land remained stable, decreased or, at least, did not increase with the same rates as those seen in the period from 1945 to 1975. The observed increased cover of natural LULC classes can be attributed to restoration projects that allowed the regrowth of natural ecosystems in deltaic areas [109] or, in some cases, due to extended salinization of areas that had been earlier delivered to agricultural exploitation following drainage projects. These fields were abandoned, and secondary vegetation succession drove LULC changes towards natural vegetation classes. Faults in the design and implementation of agricultural irrigation projects following large-scale flood control and wetland drainage works have led to increased soil salinization in many Greek wetlands [110]. Water over-abstraction and other anthropogenic pressures combined with global warming have been identified as the key reasons for the global increase of this phenomenon [111]. Since dam construction and extensive drainage are among the major large-scale interventions that prolong the dry period in wetlands and modify groundwater levels [112] it becomes clear why soil salinization is more frequent and intense in arid regions [111,113].

## 6. Conclusions

Our results demonstrate that, since 1945 in Greece, all major deltaic areas have exhibited strong simplification and homogenization trends of the structure and composition of their landscapes. Within and among deltaic areas, landscape variability decreased and nowadays, human-induced LULC classes clearly dominate the landscape at the expense of natural LULC classes as an outcome of intensification of human intervention. Patterns of change in Greek wetlands are consistent with respective patterns that have been identified in various areas of the Mediterranean and Europe, and similarly, are led by socio-economic developments. The technological availability and the need for arable land after WWII coincided with the overexploitation and the large construction projects aiming to increase productive agricultural land. Then, during the 1980s the ecological importance of wetlands was anticipated and concerns regarding their conservation emerged. Unfortunately, delayed effects of the previous intensive exploitation (e.g., the effects in coastal erosion from the construction of large dams and the arrangement of rivers) made their appearance and, in the lack of external drivers, similar landscape composition (dominated by LULC classes that are related to human activities) is expected to occur in the future. We conclude that the development of the landscape of wetland ecosystems depends mainly on social circumstances and political decisions. The study of the temporal changes improves the level of understanding regarding the long-term relationship between people and the environment and is a valuable tool for measuring the effects of management decisions. Future research with more

recent data should aim at evaluating the success of the conservation measures that have been taken during the last years, especially after the designation of the major wetlands as protected sites under the Natura 2000 directives.

**Supplementary Materials:** The following are available online at http://www.mdpi.com/2072-4292/12/8/1241/s1, Supplementary Material contains the deltaic area-based principle outputs for all delta: LULC thematic maps, changes, LULC composition, selectivity of changes, landscape metrics, and transition matrices.

**Author Contributions:** Conceptualization, N.K.; methodology, A.K., F.X., K.K., and N.K.; software, A.K.; formal analysis, A.K., F.X., and K.K.; writing—original draft preparation, A.K.; writing—review and editing, A.K., F.X., K.K., and N.K.; visualization, A.K. and N.K. All authors have read and agree to the published version of the manuscript.

**Funding:** This research received no external funding.

**Conflicts of Interest:** The authors declare no conflicts of interest.

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
