# Peer review of "Monitoring and Projecting Land Use/Land Cover Changes of Eleven Large Deltaic Areas in Greece from 1945 Onwards"

_remotesensing, doi:10.3390/rs12081241_

Round 1
Reviewer 1 Report
ID: remotesensing-764056
The manuscript "Monitoring and projecting land use/land cover changes of eleven large deltaic areas in Greece from 1945 onwards" aims to study the LULC changes from 1945 into eleven large wetlands in Greece. The areas are compared and classify based on their characteristics and the changes are project to future.
The temporal and spatial patterns of land use/land cover (LULC) were analyzed based on thematic maps generated from aerial orthophotos taken in 1945, 1975 and 2007 and a robust statistical analysis has been made to demonstrate the different patterns of landscape changes.
In general, this study has a clear research object, and a reasonable experimental design, and detailed test methods described which made its results could be trusted.
The authors have carried out a well-planned study . The objectives of the work were met. The set of data presented is quite enough and the associated literature was critically explored to support the authors' hypotheses. The text is clearly written
This paper can be accepted for pubblication for Remote Sensing
Author Response
We would like to thank the reviewer 1 for his comments.
Reviewer 2 Report
This article was well written in general and addressed spatio-temporal dynamics of land use and land cover changes of Eleven Large Deltaic Areas in Greece, from 1945 to 2007. The aim of this study is clear. It presented a clear workflow for the classification, validation, and LULC analysis. As a case study of land use/land cover change, it stands well in methodology, results, and discussion. The remote sensing part is well executed and solid.
Minor comments:
Lines 79-85 - The introduction should be improved; the aim of this study is not clear. It should define the purpose of the work and its significance, including specific hypotheses being tested. I recommend the authors to be more explicit; what are the main aims and synthesize the used methods and analysis highlighting the main conclusions. Keep the introduction comprehensible to scientists working outside the topic of the paper.
Line 109 - 10.000 ha or 10,000 ha?
Line 117 - Figure 2, remove the word legend, hectares are represented by ha and not Ha
Line 120 - How many control points by photo? 0.75 error was the average or by photo? the points have a good spatial distribution over the photos?
Line 129 - ... "RMS error of less than 0.75" meters?
Line 177 - If we have polygons generated in recognition why use "expressed by pixel number "?
Line 306 – Problem with reference "Error! Reference source not found. "
Line 387 - Figure 8, please remove the title in the figure
Lines 408-533 – In the Discussion section, it will be better to divide them into subsections. It is well written but would be better for the readers have it by subsections. Also, I would like to see more the findings and their implications in the broadest context possible and limitations of the work highlighted. It is also essential to discuss the strengths and limitations of one’s study. Comments on sources of uncertainty and error are appropriate for most papers. Future research directions should also be highlighted.
One reference example which might be useful to include: https://www.sciencedirect.com/science/article/pii/S0272771416302748
Author Response
Thank you the reviewer for his comments. We have implemented the majority of them as seen below.
Minor comments:
Lines 79-85 - The introduction should be improved; the aim of this study is not clear. It should define the purpose of the work and its significance, including specific hypotheses being tested. I recommend the authors to be more explicit; what are the main aims and synthesize the used methods and analysis highlighting the main conclusions. Keep the introduction comprehensible to scientists working outside the topic of the paper.
Reply: We decided to keep it is as small as possible without losing the purpose of the introduction.
Line 109 - 10.000 ha or 10,000 ha?
Reply: It is 10,000 corrected
Line 117 - Figure 2, remove the word legend, hectares are represented by ha and not Ha
Reply: Done
Line 120 - How many control points by photo? 0.75 error was the average or by photo? the points have a good spatial distribution over the photos?
Reply: changed to with an RMS error of less than 0.75m for all cases (ranged from 0.48m to 0.75m for each case). The number of ground control points for each case ranged from 13 to 38 having a good spatial distribution over each photo.
Line 129 - ... "RMS error of less than 0.75" meters?
Reply: Yes, please see above
Line 177 - If we have polygons generated in recognition why use "expressed by pixel number "?
Reply: All the subsequent analysis has been done considering them as rasters.
Line 306 – Problem with reference "Error! Reference source not found. "
Reply: Changed to Results of the cluster analysis are presented in Figure 6.
Line 387 - Figure 8, please remove the title in the figure
Reply: We did not understand what to change.
Lines 408-533 – In the Discussion section, it will be better to divide them into subsections. It is well written but would be better for the readers have it by subsections. Also, I would like to see more the findings and their implications in the broadest context possible and limitations of the work highlighted. It is also essential to discuss the strengths and limitations of one’s study. Comments on sources of uncertainty and error are appropriate for most papers. Future research directions should also be highlighted.
Reply: Since the paper it is not that big we would like from the reviewer to accept it as is just for the continuity of the discussion.
One reference example which might be useful to include: https://www.sciencedirect.com/science/article/pii/S0272771416302748
Reply: Done
Reviewer 3 Report
The manuscript deals with aspects of providing global patterns of LULC changes suitable for national-scaled strategic planning. The manuscript is interesting to read and adds to knowledge in this aspect.
However, the manuscript has very little to do with “remote sensing” (other than photo interpretation of images from 3 times), and may not interest the readership of “Remote Sensing”.
Accordingly, and considering the scope and the technical aspects of this manuscript, I strongly recommend that it be transferred/submitted (after corrections) to MDPI journals of Sustainability or Land or ISPRS International Journal of Geo-Informatics.
General Conceptual comments:
The authors stated that “output generalization allows for the identification of ‘global’ patterns that can provide insights suitable for the elaboration of national-scaled strategic planning” (lines 422-424). However, each deltaic area has its own socio-economic specific characteristic and development history, as shown in few places in the manuscript. Considering this fact, the authors need to support their approach for “global” instead of “local/regional” analysis of wetland, and the wisdom of doing “global” analysis within the practice of national planning, especially that their analysis did not produce an exclusive result that differ from the general/mainstream expectations.
Although I don’t doubt their validity, but the authors did not explain the need and the role of “Cluster Analysis” (lines 191-202) and “Monte Carlo Randomization” (lines 220 -231). Personally, I have never seen a study that would use these techniques in the analysis of LULC changes. I can be wrong, but I wish that the authors had elaborated on the use of these techniques as well as relate their discussion to previous studies. The results of these techniques and their interpretation were discussed in the sections of Results and Discussion, but I am not yet convinced on the need for such tasks, and what is their impact/implication on the analysis of LULC changes.
Specific comments on the Methodology:
In section 3.1, it was stated that “All images were spatially resampled to 4 m” (lines 134/135), and also that “This pixel size corresponds to the largest pixel size value among all available rasters”. However, in section 3.4.2 the images of 1975 were not used in the landscape analysis because “the resolution (scale) of these images was much lower than the other data” (lines 205/206). Furthermore, “Landscape metrics were calculated using raster format data of 5 m resolution” (lines 207/208). These sentences don’t add up, and confuse the reader. Specifically, if the largest pixel size among all available rasters is 4m, and the analysis was done using 5m resolution, why then the images of 1975 were not used?
In section 3.3, there is no discussion or assessment of the accuracy for the produced LULC maps. Normally, accuracy assessment must be done at this stage in order to ensure correct subsequent analysis.
In section 3.4.4, “the distributions of nine LULC categories were examined” (line 237) although the study used a LULC scheme of 10 categories (lines 143/144). Which category was excluded from the analysis? And why?
In section 4.1, Figure 5, “persistence” was never explained or defined (mathematically); same in Table 1.
Structure of the manuscript:
Lines 54 to 71 are redundant, not related to the topic/study area of the manuscript, and don’t add value to the discussion. I recommend removing any discussion or reference to the European situation in the manuscript. Update the list of references accordingly.
Lines 328 to 342: there is a confusion regarding what are “group 1” and “group 2”, are they “more wetland” and “more terrestrial” respectively? Or the other way around? The whole paragraph (and figure 7) needs a better restructuring and more clarifications on what constitute group 1 and group 2, and what is the meaning of the provided percentages.
Lines 425 to 454 are not related to the topic/study area of the manuscript, and don’t add value to the discussion for the readers of the journal “Remote Sensing”. As before, I recommend removing any discussion or reference to the European situation in the manuscript. Update the list of references accordingly.
Typographical error:
Lines 33-34: Incorrect and uninformative Keywords: Wetlands; Landscape (Metrics); Object-based classification; Aerial Photos (Interpretation); Mediterranean; 33 Markov (Chain); Monte Carlo (Randomization)
Line 43: insert “the” before “rivers”. Still the phrase “Human activities in rivers” sounds very weird.
Line 44: “changed the time” has no meaning here.
Line 49: “of the fourteen deltaic areas” is very vague as there is no explanation on what 14 areas and where.
Line 50: “In China, wetlands reduced by 33%” is a weak phrase. Do the authors mean wetland size/area or wetland numbers?
Line 83: “and project them to future” weak phrase.
Line 95: There is no information regarding the scale of the images or if they are all presented at the same scale.
Line 109: correct “10.000 ha” into “10,000 ha”
Line 117: The location of Kalamas is not identified on the inserted map of Greece, thus the map is redundant in this figure.
Line 124: correct “aerial photos” to “satellite images” since KH-9 is a satellite system and not an airplane.
Line 130: “united” has no meaning in “corners of the united oriented image” unless the authors meant something else and thus they should use a different word.
Line 145 and 153: “classification scheme” should be corrected to “interpretation keys”
Line 148: change “aerial photographs” into “images” or “geo-images” since they include a satellite image.
Line 162: change “and” into “that” in “segmentation process and include”
Line 167: “quality of information” is vague, do the authors mean “image resolution” or “clarity of details”?
Line 177: Change “expressed by pixel number” into “expressed by number of pixels”
Line 180: “changes tabulate matrices were estimated” is very weak and vague phrase.
Line 184: define/explain what is meant by “D” and “S” in the last column of the table.
Line 185: Change “rate” into “amount”
Lines 232-258 (Section 3.4.4): The authors kept switching between “Markove analysis” “Markove model”, and “Markov chain”. Choose one form and maintain consistency in its use.
Line 250: delete the “)” at the end of the line.
Line 306: correct the reference to Figure 6
Line 306/307: correct “t he” into “the”
Author Response
We would like to thank the reviewer for the valuable comments made. Most of them we have implemented.
Comments and Suggestions for Authors
The manuscript deals with aspects of providing global patterns of LULC changes suitable for national-scaled strategic planning. The manuscript is interesting to read and adds to knowledge in this aspect.
However, the manuscript has very little to do with “remote sensing” (other than photo interpretation of images from 3 times), and may not interest the readership of “Remote Sensing”.
Accordingly, and considering the scope and the technical aspects of this manuscript, I strongly recommend that it be transferred/submitted (after corrections) to MDPI journals of Sustainability or Land or ISPRS International Journal of Geo-Informatics.
General Conceptual comments:
The authors stated that “output generalization allows for the identification of ‘global’ patterns that can provide insights suitable for the elaboration of national-scaled strategic planning” (lines 422-424). However, each deltaic area has its own socio-economic specific characteristic and development history, as shown in few places in the manuscript. Considering this fact, the authors need to support their approach for “global” instead of “local/regional” analysis of wetland, and the wisdom of doing “global” analysis within the practice of national planning, especially that their analysis did not produce an exclusive result that differ from the general/mainstream expectations.
Although I don’t doubt their validity, but the authors did not explain the need and the role of “Cluster Analysis” (lines 191-202) and “Monte Carlo Randomization” (lines 220 -231). Personally, I have never seen a study that would use these techniques in the analysis of LULC changes. I can be wrong, but I wish that the authors had elaborated on the use of these techniques as well as relate their discussion to previous studies. The results of these techniques and their interpretation were discussed in the sections of Results and Discussion, but I am not yet convinced on the need for such tasks, and what is their impact/implication on the analysis of LULC changes.
Reply: The cluster analysis helped us to group the changes and discover that despite the small differences these can be grouped in two main classes which denotes also the “global” character. The Monte Carlo helped us to check if the changes are statistically significant and if they are more or less than those expected by chance which is a nice, we believe, approach. This gives us the knowledge to characterize more in depth the changes
Specific comments on the Methodology:
In section 3.1, it was stated that “All images were spatially resampled to 4 m” (lines 134/135), and also that “This pixel size corresponds to the largest pixel size value among all available rasters”. However, in section 3.4.2 the images of 1975 were not used in the landscape analysis because “the resolution (scale) of these images was much lower than the other data” (lines 205/206). Furthermore, “Landscape metrics were calculated using raster format data of 5 m resolution” (lines 207/208). These sentences don’t add up, and confuse the reader. Specifically, if the largest pixel size among all available rasters is 4m, and the analysis was done using 5m resolution, why then the images of 1975 were not used?
Reply: We decided not to use the 1975, in the landscape analysis because these images were not that detailed, and we thought that the landscape metrics will be influenced although we could set up the raster resolution to 4 or 5 meters
In section 3.3, there is no discussion or assessment of the accuracy for the produced LULC maps. Normally, accuracy assessment must be done at this stage in order to ensure correct subsequent analysis.
Reply: Here there is an issue on how to estimate the accuracy especially when the images refer to the very past.
In section 3.4.4, “the distributions of nine LULC categories were examined” (line 237) although the study used a LULC scheme of 10 categories (lines 143/144). Which category was excluded from the analysis? And why?
Reply: There is a mistake in the numbering of the classes in the lines 143/144. (1) Urban areas, (2) Agricultural areas, (3) Wetlands, (4) Inland water, (6) Bare land, (7) Grasslands, (8) Shrublands, (9) Forests, and (10) Sea. From 4 we go to 6, therefore the categories are 9, we changed it.
In section 4.1, Figure 5, “persistence” was never explained or defined (mathematically); same in Table 1.
Reply: I table 1 it is written Diagonal elements show the percentage of persistence of the class
Structure of the manuscript:
Lines 54 to 71 are redundant, not related to the topic/study area of the manuscript, and don’t add value to the discussion. I recommend removing any discussion or reference to the European situation in the manuscript. Update the list of references accordingly.
Reply: Since the study area belongs to Europe, we thought that it will be interesting to see what is happening in Europe, which seems that similar changes are occurring. If the reviewer insists, we could modify in another revision.
Lines 328 to 342: there is a confusion regarding what are “group 1” and “group 2”, are they “more wetland” and “more terrestrial” respectively? Or the other way around? The whole paragraph (and figure 7) needs a better restructuring and more clarifications on what constitute group 1 and group 2, and what is the meaning of the provided percentages.
Reply: We clarified which group goes to which classes. “landscapes into "more wetland (group 2)" or "more terrestrial (group 1)" ecosystems already from 1945 (Figure 7).” Also, it is clear in the next sentence “In 1945 anthropogenic ecosystems occupied areas of about 40% and 33% in groups 1 and 2, respectively, wetlands occupied 22% and 53%, and terrestrial ecosystems occupied 38% and 13% in groups 1 and 2, respectively.”
Lines 425 to 454 are not related to the topic/study area of the manuscript, and don’t add value to the discussion for the readers of the journal “Remote Sensing”. As before, I recommend removing any discussion or reference to the European situation in the manuscript. Update the list of references accordingly.
Reply: Same as above Since the study area belongs to Europe, we thought that it will be interesting to see what is happening in Europe, which seems that similar changes are occurring. If the reviewer insists, we could modify in another revision.
Typographical error:
Lines 33-34: Incorrect and uninformative Keywords: Wetlands; Landscape (Metrics); Object-based classification; Aerial Photos (Interpretation); Mediterranean; 33 Markov (Chain); Monte Carlo (Randomization)
Reply: We changed them as suggested
Line 43: insert “the” before “rivers”. Still the phrase “Human activities in rivers” sounds very weird.
Reply: We added it
Line 44: “changed the time” has no meaning here.
Reply: We deleted “time”
Line 49: “of the fourteen deltaic areas” is very vague as there is no explanation on what 14 areas and where.
Reply: We deleted the “the”
Line 50: “In China, wetlands reduced by 33%” is a weak phrase. Do the authors mean wetland size/area or wetland numbers?
Reply: wetland area. We changed it accordingly
Line 83: “and project them to future” weak phrase.
Reply: We changed to “project their synthesis to future”
Line 95: There is no information regarding the scale of the images or if they are all presented at the same scale.
Reply: It is not indicated the scale to avoid too many numbers, but the box is 10000 ha, therefore there is an indication about t he scale.
Line 109: correct “10.000 ha” into “10,000 ha”
Reply: Done
Line 117: The location of Kalamas is not identified on the inserted map of Greece, thus the map is redundant in this figure.
Line 124: correct “aerial photos” to “satellite images” since KH-9 is a satellite system and not an airplane.
Reply: We changed the title to Aerial and satellite photographs
Line 130: “united” has no meaning in “corners of the united oriented image” unless the authors meant something else and thus they should use a different word.
Reply: We deleted the word “united
Line 145 and 153: “classification scheme” should be corrected to “interpretation keys”
Reply: We mean actually the classification legend, interpretations keys we believe it is something else.
Line 148: change “aerial photographs” into “images” or “geo-images” since they include a satellite image.
Reply: Done we used the “geo-images”
Line 162: change “and” into “that” in “segmentation process and include”
Reply: Done
Line 167: “quality of information” is vague, do the authors mean “image resolution” or “clarity of details”?
Reply: We changed it to “best image resolution and quality”
Line 177: Change “expressed by pixel number” into “expressed by number of pixels”
Reply: Done
Line 180: “changes tabulate matrices were estimated” is very weak and vague phrase.
Reply: We changed it to “cross-tabulation matrices”
Line 184: define/explain what is meant by “D” and “S” in the last column of the table.
Reply: We added D Net Change and S Swap
Line 185: Change “rate” into “amount”
Reply: Done
Lines 232-258 (Section 3.4.4): The authors kept switching between “Markove analysis” “Markove model”, and “Markov chain”. Choose one form and maintain consistency in its use.
Reply: We checked it and we use all time the Markov, except once where we use the Markovian
Line 250: delete the “)” at the end of the line.
Reply: Done
Line 306: correct the reference to Figure 6
Reply: Done
Line 306/307: correct “t he” into “the”
Reply: It seems it is corrcted